# Study on Single Nucleotide Polymorphism of *LAP3* Gene and Its Correlation with Dairy Quality Traits of Gannan Yak

**DOI:** 10.3390/foods13182953

**Published:** 2024-09-18

**Authors:** Tong Wang, Xiaoming Ma, Fen Feng, Fei Zheng, Qingbo Zheng, Juanxiang Zhang, Minghao Zhang, Chaofan Ma, Jingying Deng, Xian Guo, Min Chu, Yongfu La, Pengjia Bao, Heping Pan, Chunnian Liang, Ping Yan

**Affiliations:** 1Key Laboratory of Yak Breeding Engineering of Gansu Province, Lanzhou Institute of Husbandry and Pharmaceutical Sciences, Chinese Academy of Agricultural Sciences, Lanzhou 730000, China; 2Key Laboratory of Animal Genetics and Breeding on Tibetan Plateau, Ministry of Agriculture and Rural Affairs, Lanzhou 730000, China; 3Life Science and Engineering College, Northwest Minzu University, Lanzhou 730124, China; 4Institute of Western Agriculture, Chinese Academy of Agricultural Sciences, Changji 931100, China

**Keywords:** Gannan yak, *LAP3* gene, SNPs, molecular marker

## Abstract

This study explored the polymorphism of the leucine aminopeptidase (*LAP3*) gene and its relationship with milk quality characteristics in Gannan yak. A cohort of 162 Gannan yak was genotyped utilizing the Illumina Yak cGPS 7K BeadChip, and the identified single nucleotide polymorphisms (SNPs) were evaluated for their association with milk protein, casein, lactose, and fat concentrations. The results showed that four SNPs (g.4494G > A, g.5919A > G, g.8033G > C, and g.15,615A > G) in the *LAP3* gene exhibited polymorphism with information content values of 0.267, 0.267, 0.293, and 0.114, respectively. All four SNPs were in Hardy–Weinberg equilibrium (*p* > 0.05). The g.4494G > A and g.5919A > G SNPs were significantly associated with protein content (*p* < 0.05), with homozygous genotypes showing significantly higher protein content than heterozygous genotypes (*p* < 0.05). The g.8033G > C SNP was significantly associated with casein content, protein content, non-fat solids, and acidity (*p* < 0.05), with the CC genotype having significantly higher casein, protein, and non-fat solids content than the GG and GC genotypes (*p* < 0.05). The g.15,615A > G SNP was significantly associated with average fat globule diameter (*p* < 0.05). In general, the mutations within the *LAP3* gene demonstrated a positive impact on milk quality traits in Gannan yak, with mutated genotypes correlating with enhanced milk quality. These results indicate that the *LAP3* gene could be a significant or candidate gene affecting milk quality traits in Gannan yak and offer potential genetic markers for molecular breeding programs in this species.

## 1. Introduction

Yak (*Bos grunniens)*, a rare species adapted to high-altitude environments above 3000 m [1], are often referred to as the “Ship of the Plateau” due to their remarkable ability to thrive in harsh conditions characterized by low temperatures, hypoxia, strong ultraviolet radiation, and low atmospheric pressure [2]. This exceptional adaptability makes yak an indispensable livestock species for local herders [3]. They provide crucial resources such as meat, milk, and leather, contributing significantly to the livelihood of the communities they inhabit. Furthermore, the yak’s significant draft power greatly facilitates the daily lives of herders, earning it the title of “All-Purpose Livestock” and cementing its unique status in the high-altitude regions [4]. The yak meat produced in the natural plateau pastures has unique characteristics; it is tender, its color is bright red, and it contains high protein, low fat, and rich minerals and vitamins [5]. Yak milk is a unique, high-quality dairy source indigenous to the plateau region. In comparison to other milks, it is notably richer in proteins, amino acids, minerals, and vitamins [6], making it highly digestible and nutritionally dense. The composition of milk plays a crucial role in determining the yield and quality of dairy products. Yak milk boasts a high casein content with a diverse casein profile, contributing to its excellent emulsifying and foaming properties [7]. Yak milk, often referred to as “naturally concentrated milk,” holds a prominent place in the daily diet of herders. As a nutritionally complete food source, it has demonstrated beneficial effects in enhancing human immunity, promoting bone density, and exhibiting strong antioxidant, cholesterol-lowering, anti-atherosclerotic, and anti-tumor properties [8]. This makes yak milk a raw material with immense potential for development and utilization. However, the yak milk industry in China remains underdeveloped. This is attributed to several factors, including the limited use of yak milk in industrial production, the relatively short lactation period of yak, and their lower milk yield due to the challenging environmental conditions they endure. Therefore, improving milk quality and yield is a pressing need and remains a key challenge for the Chinese dairy industry.

Traditional breeding methods for improving milk protein percentage are not only time-consuming but also inefficient. With the rapid advancements in molecular biology, breeding technologies have entered a phase where traditional and molecular breeding approaches are integrated. Molecular breeding provides substantial benefits for traits including milk yield, fat content, and protein levels. Currently, widely researched molecular breeding methods primarily include molecular marker-assisted selection (MAS), genome-wide association study (GWAS), and genotyping through the targeted sequencing of liquid-captured fragments (cGPS) [9]. Single nucleotide polymorphisms (SNPs), after restriction fragment length polymorphisms (RFLP) and simple sequence repeats (SSR), represent the third generation of molecular marker technology [10]. Studies have revealed the presence of nearly 80 million SNPs in bovine genomes and 3–10 million SNPs in the human genome, with an estimated occurrence of one SNP per 1000 base pairs. Numerous studies have concentrated on utilizing methods such as genome-wide association studies (GWASs) or quantitative trait loci (QTL) analysis to identify genes linked with specific traits. Ma et al. [11] genotyped 172 yak and identified single nucleotide polymorphisms (SNPs) in the *PRKD1* and *KCNQ3* genes that were significantly associated with yak milk quality traits. Shymaa et al. [12] identified four exonic mutations in the growth hormone receptor (*GHR*) gene in 400 Egyptian buffalo. They found that selecting buffalo carrying the AA haplotype for this gene led to improved milk yield and quality. Abousoliman et al. [13] conducted a genome-wide association study in high-performing and low-performing ewes, identifying candidate genes associated with milk yield (*SLC4A8*, *NUB1*, and *TBC1D1*) and milk quality traits (*PPARA* and *FBLN1*). Yak milk production is influenced by many factors, and studies have shown that many genes (*LAP3*, *CSN1S1*, *DGAT1* and *RPL8*) are related to milk production traits. The influence of these genes on the proteins in milk is well known [14].

Leucine aminopeptidase 3 (*LAP3*), an exopeptidase, is an enzyme responsible for catalyzing the hydrolytic cleavage of leucine from the N-termini of peptides [15]. This enzyme exhibits substrate specificity, promoting and prolonging the hydrolysis of leucine under optimal pH and temperature conditions. *LAP3* is widely distributed throughout the animal body, with predominant expression in the liver, pancreas, and kidneys. It plays a key role in the degradation of tissue proteins and peptides. In humans, *LAP3* is located on chromosome 4 and encodes 519 amino acids. In cattle, *LAP3* resides on chromosome 6 and also encodes 519 amino acids. Similarly, *LAP3* is located on chromosome 6 in sheep. Sheehy et al. [16] observed significant variations in *LAP3* expression across different stages of the lactation cycle, with peak expression levels during lactation, reaching 2.2 times the levels observed during gestation. Cohen-Zinder et al. [17] found that different genotypes of *LAP3* in Hereford cattle resulted in variations in milk protein and fat content. Hawlader et al. [18] established a significant correlation between *LAP3* and body weight in Australian Merino sheep. Hanne et al. [19] identified four polymorphic sites within the *LAP3* gene sequence and calculated their allele frequencies. Gannan yak, recognized as a valuable genetic resource in China, exhibit strong resilience. This study focuses on Gannan yak as the experimental model to investigate the genetic polymorphism of the *LAP3* gene and its association with yak milk quality traits. The findings of this research aim to provide valuable data for improving yak milk quality and developing molecular markers for the yak population.

## 2. Materials and Methods

### 2.1. Ethics Statement

All animal experiments carried out in this study received approval from the Animal Ethics Committee of the Institute of Animal Science and Veterinary Medicine, Chinese Academy of Agricultural Sciences, Lanzhou, with authorization number 1610322020018.

### 2.2. Experimental Animal Selection

In this study, 162 Gannan yak cows from Xiahe County, Gannan Tibetan Autonomous Prefecture, Gansu Province (altitude ranging from 3000 to 3800 m) were selected as experimental subjects. All animals were in their lactation period and had calved 2 to 3 times. Samples were collected on the same day. The selected yak cows exhibited similar body conditions and comparable milk yields. These 162 Gannan yak cows had been monitored by the local animal husbandry station over an extended period and were confirmed to be free from mastitis and reproductive-related diseases. They were grazing on the same natural pasture in the Gannan region and were managed under a fully pastoral system without any artificial feed supplementation.

### 2.3. Sample Collection and DNA Extraction

Tissue samples from the ear and milk were collected from 162 Gannan yak for DNA extraction and an analysis of milk components, respectively. Collected ear tissue samples were immediately immersed in liquid nitrogen for temporary preservation and subsequently stored at −80 °C in the laboratory. DNA extraction was conducted using the MGIEasy Genomic DNA Extraction Kit (V1.0, 48Preps, Shenzhen, China), (magnetic bead method) according to the manufacturer’s instructions. The DNA concentration was measured using a Qubit fluorometer 4.0 (Thermo Field, Nashville, TN, USA) [20], and DNA integrity was assessed via 1% agarose (Yeasen, Shanghai, China) gel electrophoresis. Samples that passed quality control were used for library construction. Milk samples were collected from lactating female yak in the morning using manual milking. Before collection, the udder area was cleaned and disinfected, followed by gentle massage stimulation. Collected milk samples were transferred into 50 mL centrifuge tubes, labeled with the ear tag number and sampling time, and stored at 4 °C prior to transportation to the laboratory for milk quality analysis.

### 2.4. Genotyping

The genotyping of the aforementioned 162 Gannan yak was performed using the Illumina Yak cGPS 7K liquid chip (Illumina, Huazhi Biotechnology Co., Ltd., Changsha, China). cGPS is a precise localization sequencing genotyping technology based on target sequence liquid phase capture. cGPS offers advantages including high information content, high detection rate, high stability, and high autonomy [21,22]. It uses synthesized specific probes to capture and enrich multiple target sequences from various genomic locations through liquid-phase hybridization. Subsequently, sequencing and library construction are performed on the targeted sequence intervals, ultimately obtaining the genotypes of marker loci within the target sequence regions. The raw sequencing reads obtained contain adapter sequences and low-quality sequences. To ensure accuracy, the raw data were subjected to quality control using Fastp (v0.23.4) [23]. Randomly selected 10,000 sequences from each sample’s fastq file were aligned to the NCBI-NT database using blastn [24] for contamination assessment. The filtered data were aligned to the yak reference genome *Bos grunniens* v3.0 (GCA_005887515.1) using the alignment software BWA (v0.7.17) [25]. The genotype detection of target loci was conducted using the HaplotypeCaller variant detection tool from the GATK (v4.1.5.0) mutation analysis software. Finally, the functional annotation of detected genetic variations was performed using the ANNOVAR [26] software, revealing the genomic regions where the variations occurred and the impact of the variations.

### 2.5. Analysis of Milk Composition of Gannan Yak

Milk samples collected from the 162 Gannan yak were analyzed for casein, protein, fat, total solids, solids-not-fat, lactose, average fat globule diameter, and acidity using a MilkoScanTM FT120 milk component analyzer (Danish FUCHS Analytical Instruments Ltd., Hellerup, Denmark).

### 2.6. Statistical Analysis of Data

The theoretical heterozygosity, observed heterozygosity, effective number of alleles, polymorphism information content, theoretical genotype frequency, genotype deviation, and Hardy–Weinberg test *p*-value for the LAP3 gene locus were calculated using the GDICALL online software (http://www.msrcall.com/gdicall.aspx, accessed on 13 August 2024) [27]. The relationship between LAP3 gene polymorphism and milk quality traits in Gannan yak was assessed using one-way analysis of variance (ANOVA) in IBM SPSS Statistics 25 (IBM, Armonk, NY, USA) [28]. Differences among means were analyzed using Duncan’s multiple range test. Results are presented as mean ± standard deviation. Significance was set at *p* < 0.05 and high significance at *p* < 0.01.

## 3. Results

### 3.1. Genotyping Results of LAP3 and Analysis of Genetic Parameters of LAP3 Gene Locus in Gannan Yak

Figure 1 shows the distribution of genotypes for different SNPs within the LAP3 gene of Gannan yak in relation to milk quality traits. The figure reveals a relatively even distribution of SNPs. The study calculated genotype frequencies, allele frequencies, theoretical heterozygosity, observed heterozygosity, and polymorphism information content (PIC) for the LAP3 gene locus in Gannan yak, as presented in Table 1. Among the genotype frequencies for LAP3 gene g.4494G > A, g.5919A > G, g.8033G > C, and g.15,615A > G, GG, AA, GG, and AA exhibited the highest frequencies, at 0.630, 0.630, 0.599, and 0.870, respectively. This indicates a predominance of homozygous genotypes at these four SNP loci. Across these four SNP loci, the allele frequencies for G, A, G, and A were 0.802, 0.802, 0.769, and 0.935, respectively, suggesting a dominant role of non-mutated alleles at these loci. Calculations revealed that observed heterozygosity generally exceeded theoretical heterozygosity in this population, indicating the presence of a higher proportion of heterozygous individuals than expected under random mating. Further calculations of PIC for the four loci g.4494G > A, g.5919A > G, g.8033G > C, and g.15,615A > G resulted in values of 0.267, 0.267, 0.293, and 0.114, respectively. These findings demonstrate the presence of genetic polymorphism at these four loci within the population, although the degree of polymorphism varies. The first three loci exhibited PIC values between 0.25 and 0.5, indicating moderate polymorphism, while the last locus displayed a PIC value below 0.25, suggesting low polymorphism. All four loci, g.4494G > A, g.5919A > G, g.8033G > C, and g.15,615A > G, were found to be in agreement with the Hardy–Weinberg equilibrium (*p* > 0.05).

### 3.2. Correlation Analysis of LAP3 Genotype and Dairy Quality Traits in Gannan Yak

The study examined the correlation between various SNPs and milk composition traits in Gannan yak, using SNP genotyping data. The results are presented in Table 2, which lists the correlations between four loci in the LAP3 gene (g.4494G > A, g.5919A > G, g.8033G > C, and g.15,615A > G) and milk quality traits in Gannan yak. The g.4494G > A locus showed a significant association with protein content (*p* < 0.05). The AA homozygous genotype showed significantly higher protein content compared to the GA genotype (*p* < 0.05), but the difference with the GG genotype was not statistically significant (*p* > 0.05). Similar results were observed at the g.5919A > G locus, which also displayed a significant association with protein content (*p* < 0.05). The GG genotype showed significantly higher protein content than the AG genotype (*p* < 0.05). However, no significant difference in protein content was observed between the GG and AA genotypes (*p* > 0.05). The g.8033G > C locus showed significant associations with casein content, protein content, solids-not-fat, and acidity (*p* < 0.05). The CC genotype exhibited significantly higher casein, protein, and solids-not-fat content compared to both the GG and GC genotypes (*p* < 0.05); no significant difference in protein content was observed between the GG and GC genotypes (*p* > 0.05). The CC genotype showed significantly higher acidity than the GC genotype (*p* < 0.05). However, no significant difference in acidity was observed between the CC and GG genotypes (*p* > 0.05). The g.15,615A > G locus showed a significant association with average fat globule diameter (*p* < 0.05). The analysis of the correlation between different genotypes of the LAP3 gene and milk composition traits in Gannan yak revealed that the LAP3 gene is primarily associated with casein, protein, solids-not-fat, acidity, and average fat globule diameter in Gannan yak milk. In the milk components significantly associated with the three loci g.4494G > A, g.5919A > G, and g.8033G > C, the mutant homozygous genotype showed significantly higher values compared to the mutant heterozygous and wild homozygous genotypes. However, at the g.15,615A > G locus, the wild homozygous genotype showed significantly higher values compared to the mutant heterozygous genotype. Overall, mutations appear to have a positive impact on milk quality in Gannan yak, with mutant yak individuals exhibiting higher milk quality.

### 3.3. SNPs Linkage Disequilibrium Analysis and KEGG Functional Enrichment Analysis of LAP3 Gene in Gannan Yak

The study utilized an online tool (https://www.bioinformatics.com.cn/, accessed on 13 August 2024) to perform linkage disequilibrium analysis on four loci (g.4494G > A, g.5919A > G, g.8033G > C, and g.15,615A > G) within the LAP3 gene of Gannan yak. The results, depicted in Figure 2, indicate a complete linkage disequilibrium between the g.4494G > A and g.5919A > G loci. Complete linkage equilibrium was observed between the g.8033G > C and g.15,615A > G loci and the g.4494G > A and g.5919A > G loci. KEGG pathway enrichment analysis of the LAP3 gene was conducted using the KOBAS tool (http://bioinfo.org/kobas/annotate/, accessed on 13 August 2024). The results revealed an enrichment of the LAP3 gene in the pathways relevant to milk quality, including arginine and proline metabolism and glutathione metabolism.

## 4. Discussion

Yak milk, a rare dairy product, originates from the Qinghai–Tibet Plateau and serves as the primary ingredient for local herders’ dairy production and a crucial source of supplementary nutrition. However, yak milk production is significantly lower compared to other milk types, with daily yields typically less than 10% of that from ordinary dairy cows. Furthermore, milk production is limited to the lactation period. Despite its lower output, yak milk boasts significantly higher nutritional value than ordinary cow milk [29]. Yak milk is rich in various nutrients, including conjugated linoleic acid, protein, lactose, calcium, lactoferrin, immunoglobulins, and unsaturated fatty acids [30]. It has been dubbed a “superfood” by the British “The Observer” and holds significant economic and nutritional value, particularly suited for consumption by individuals with higher nutritional needs, such as lactating infants, the elderly, and those with weakened immune systems [31]. Protein is a major nutritional component of milk, with significantly higher content in yak milk compared to ordinary cow milk. [11] Experimental results indicated that the g.4494G > A and g.5919A > G loci primarily influence protein content in Gannan yak milk, with homozygous AA and GG genotypes showing significantly higher protein levels compared to heterozygous genotypes. The g.8033G > C locus primarily affects protein and casein content in Gannan yak milk, with homozygous CC exhibiting significantly higher levels than both heterozygous genotypes. Milk protein comprises two main categories: casein and whey protein. Casein, a phosphorus-containing protein, contains all eight essential amino acids for humans. It is the most abundant protein in milk, reaching 1.5 times the amount found in ordinary cow milk and 11 times that of human milk [32]. Studies have demonstrated that the composition and content of various casein proteins in milk significantly influence its renneting properties [33]. As dairy science research progresses, attention towards milk and dairy products has shifted from merely focusing on the quantity of nutritional components to a greater emphasis on their quality. Research suggests a strong correlation between the characteristic aroma of yak milk and its high fat content, with higher fat content resulting in a more pronounced aroma [34]. Yak milk is higher in fat content, with a greater proportion of saturated fatty acids and a lower content of unsaturated fatty acids compared to other milk types [35]. These differences contribute to the unique flavor profile of yak milk. Research has also shown that flavor plays a crucial role in determining the sensory value of dairy products and influences the assessment of food freshness and nutritional value [36]. The findings of this study indicate that SNP mutations can significantly enhance milk quality in Gannan yak.

Leucine aminopeptidase (LAP), belonging to the esteemed M17 aminopeptidase family, was originally identified in bovine lens tissue. Its designation stems from its unique capability to precisely excise leucine residues from shorter peptide chains, a process facilitated by a distinctive binuclear metal center, thereby showcasing its specialized enzymatic function [15]. The M17 aminopeptidase family comprises numerous members, most of which share a characteristic sequence, NXDAEGRL [37]. LAP is expressed in various tissues, including the lens, kidney, and mammary gland, and it interacts with DNA [38]. It is also implicated in biological processes such as cancer cell proliferation and angiogenesis [39,40]. *LAP3*, another member of the M17 aminopeptidase family, was initially discovered alongside LAP4 [41]. It occupies a pivotal position in the intricate processes of protein degradation and the intricate metabolism of bioactive peptides, playing a vital role in regulating their turnover and functional activity. [42,43]. This study identified four SNPs within the *LAP3* gene: g.4494G > A, g.5919A > G, g.8033G > C, and g.15,615A > G, which are associated with milk quality traits in Gannan yak. All four SNPs are located within introns. While introns were long considered to be sequences without a definite function, essentially “junk” DNA, a growing body of research is increasingly recognizing their significant biological roles [44]. Introns can influence gene expression at various levels, including transcription, polyadenylation, and translation efficiency, through the facilitation of alternative splicing; they enable the diverse generation of multiple protein isoforms from a solitary gene. Numerous studies have demonstrated that introns are essential for optimal gene expression [45]. Genes with 5′ introns containing regulatory elements exhibit increased expression compared to those with other introns [44]. Therefore, the specific mechanisms by which the four identified SNPs within the *LAP3* gene influence milk quality traits in Gannan yak warrant further investigation. Zheng et al. [46] discovered that the *LAP3* gene influences milk yield, milk fat percentage, and milk protein percentage in cattle. Similarly, Wei et al. [47] identified a correlation between the *LAP3* gene and lactation performance in sheep through genome-wide association analysis. Worku et al. [48] performed a correlation analysis of milk production traits in 263 dairy cows (Sahiwal and Karan Fries) and found that the SNPs rs110,839,532: G > T, rs43,702,361: T > C, and rs41,255,599: C > T, located within the 3′UTR of the *LAP3* gene, were significantly associated with 305-day milk yield and lactation duration, suggesting a potential role of *LAP3* gene variation in these traits. These findings align with the current study, which revealed that the four identified SNPs exhibited polymorphism information content values of 0.267, 0.267, 0.293, and 0.114, respectively, and all adhered to the Hardy–Weinberg equilibrium. Based on these observations, the study proposes that the *LAP3* gene could serve as a molecular marker, providing a theoretical foundation for yak breeding programs.

## 5. Conclusions

This research marks the initial exploration of the linkage between *LAP3* gene polymorphisms and milk quality attributes in Gannan yak, revealing a notable association between four specific SNPs within the *LAP3* gene and these valuable traits. These identified SNPs hold potential as informative molecular markers for milk quality in yak, offering invaluable perspectives for advancing molecular breeding strategies and fostering the creation of novel yak breeds with enhanced milk quality.

## Figures and Tables

**Figure 1 foods-13-02953-f001:**
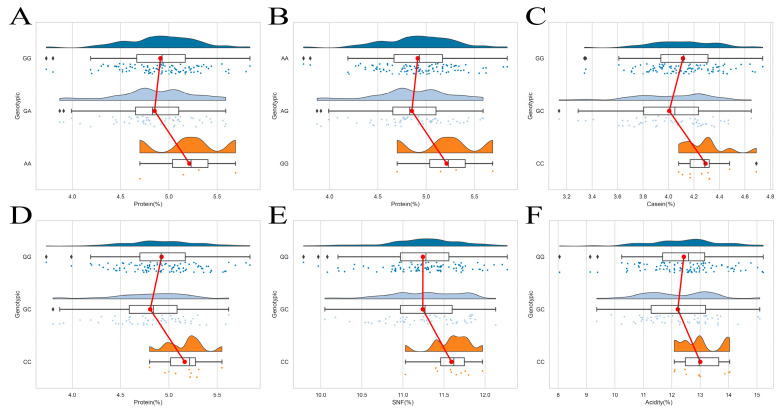
Distribution of SNP genotypes. (**A**): g.4494G > A; (**B**): g.5919A > G; (**C**–**F**): g.8033G > C. The three different colored dots in the figure represent the distribution of genotype data, and the red line indicates the average value of the milk quality trait associated with the genotype.

**Figure 2 foods-13-02953-f002:**
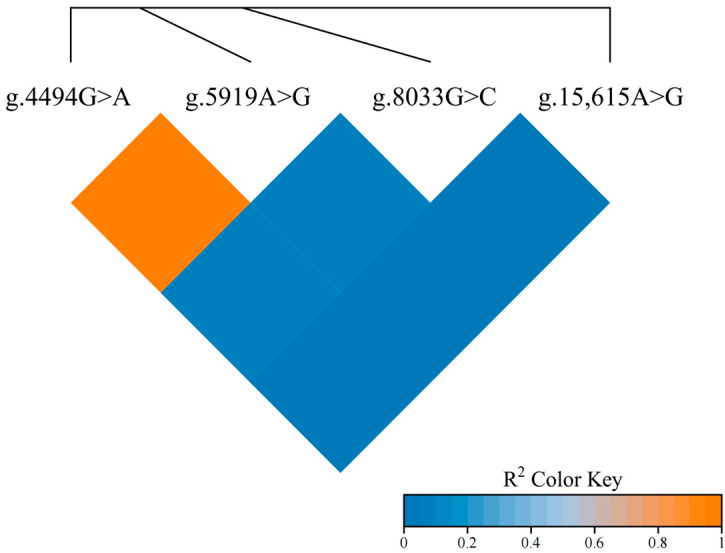
Linkage disequilibrium analysis among the four SNPs of LAP3 gene.

**Table 1 foods-13-02953-t001:** Variation information and diversity parameters of *LAP3* gene locus.

SNPs	Position	Genotypic Frequencies				Allelic Frequencies		He	Ho	PIC	HW *p* Value
g.4494G > A	Intron	GG	GA		AA	G	A	0.317	0.683	0.267	0.250
		0.630	0.346		0.025	0.802	0.198				
g.5919A > G	Intron	AA	AG		GG	A	G	0.317	0.683	0.267	0.250
		0.630	0.346		0.025	0.802	0.198				
g.8033G > C	Intron	GG	GC		CC	G	C	0.356	0.644	0.293	0.560
		0.599	0.340		0.062	0.769	0.231				
g.15,615A > G	Intron	AA		AG		A	G	0.121	0.879	0.114	0.378
		0.870		0.130		0.935	0.065				

Note: He: heterozygosity; Ho: homozygosity; polymorphism information content (PIC) values less than 0.25 indicate low polymorphism, values between 0.25 and 0.5 indicate moderate polymorphism, and values greater than 0.5 indicate high polymorphism. A HW *p* value > 0.05 indicates that the population’s genes are in Hardy–Weinberg equilibrium, suggesting that the samples originated from the same Mendelian population.

**Table 2 foods-13-02953-t002:** Correlation analysis of *LAP3* gene G.4494 G > A, G.5919 A > G, G.8033 g > C and G.15,615 A > G with dairy quality traits in Gannan yak.

g.4494G > A								
Genotype	Casein/%	Protein/%	Fat/%	TS/%	SNF/%	Lactose/%	FPD/μm	Acidity/°T
GG	4.11 ± 0.28	4.91 ± 0.39 ^ab^	5.64 ± 2.67	16.81 ± 2.56	11.30 ± 0.49	5.00 ± 0.15	0.71 ± 0.27	12.38 ± 1.38
GA	4.04 ± 0.32	4.85 ± 0.41 ^b^	5.52 ± 2.61	16.60 ± 2.67	11.18 ± 0.45	4.96 ± 0.16	0.71 ± 0.30	12.35 ± 1.12
AA	4.22 ± 0.17	5.21 ± 0.41^a^	4.18 ± 2.69	15.57 ± 2.43	11.54 ± 0.31	4.87 ± 0.28	0.70 ± 0.00	13.33 ± 1.11
**g.5919A > G**								
**Genotype**	**Casein/%**	**Protein/%**	**Fat/%**	**TS/%**	**SNF/%**	**Lactose/%**	**FPD/μm**	**Acidity/°T**
AA	4.11 ± 0.28	4.91 ± 0.39 ^ab^	5.64 ± 2.67	16.81 ± 2.56	11.30 ± 0.49	5.00 ± 0.15	0.71 ± 0.27	12.38 ± 1.38
AG	4.04 ± 0.32	4.85 ± 0.41 ^b^	5.52 ± 2.61	16.60 ± 2.67	11.18 ± 0.45	4.96 ± 0.16	0.71 ± 0.30	12.35 ± 1.12
GG	4.22 ± 0.17	5.21 ± 0.41 ^a^	4.18 ± 2.69	15.57 ± 2.43	11.54 ± 0.31	4.87 ± 0.28	0.70 ± 0.00	13.33 ± 1.11
**g.8033G > C**								
**Genotype**	**Casein/%**	**Protein/%**	**Fat/%**	**TS/%**	**SNF/%**	**Lactose/%**	**FPD/μm**	**Acidity/°T**
GG	4.11 ± 0.27^b^	4.92 ± 0.39 ^b^	5.84 ± 2.90	16.98 ± 2.77	11.24 ± 0.49 ^b^	4.96 ± 0.15	0.71 ± 0.28	12.43 ± 1.34 ^ab^
GC	4.00 ± 0.32^b^	4.81 ± 0.40 ^b^	5.06 ± 2.18	16.16 ± 2.23	11.24 ± 0.46 ^b^	5.02 ± 0.16	0.70 ± 0.27	12.21 ± 1.26 ^b^
CC	4.29 ± 0.18^a^	5.16 ± 0.21^a^	5.64 ± 2.27	17.09 ± 2.31	11.59 ± 0.26 ^a^	5.02 ± 0.12	0.71 ± 0.32	13.00 ± 0.74 ^a^
**g.15,615A > G**								
**Genotype**	**Casein/%**	**Protein/%**	**Fat/%**	**TS/%**	**SNF/%**	**Lactose/%**	**FPD/μm**	**Acidity/°T**
AA	4.10 ± 0.30	4.91 ± 0.41	5.62 ± 2.73	16.77 ± 2.67	11.27 ± 0.48	4.98 ± 0.15	0.71 ± 0.03 ^a^	12.42 ± 1.32
AG	4.01 ± 0.20	4.80 ± 0.29	5.19 ± 2.05	16.31 ± 1.92	11.23 ± 0.41	5.04 ± 0.15	0.70 ± 0.00 ^b^	12.21 ± 1.10

Note: Different lowercase letters within the same dataset indicate statistically significant differences (*p* < 0.05). Data are presented as mean ± standard deviation.

## Data Availability

The original contributions presented in the study are included in the article; further inquiries can be directed to the corresponding authors.

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
