# Peer review of "Study on Single Nucleotide Polymorphism of LAP3 Gene and Its Correlation with Dairy Quality Traits of Gannan Yak"

_foods, 2024, doi:10.3390/foods13182953_

Round 1

Reviewer 1 Report

Comments and Suggestions for Authors

Leucine aminopeptidases (LAPs) is exopeptidase which catalyses the removal of N-terminal amino acids and is a part of family of aminopeptidases that have been found in many tissues including lens, kidney, pancreas, muscle, liver, mammary and subcellular locations in a diverse of species. LAPs are often viewed as cell maintenance enzymes with critical roles in turnover of peptides. In mammals, LAPs contribute to processing of bioactive peptides (oxytocin, vasopressin, enkephalins), and vesicle trafficking to the plasma membrane and have a role in MHC I antigen presentation. Sera from pregnant women contain two kinds of LAP activity, one is normal LAP, the other is placental LAP (P-LAP), which appears only during pregnancy and was found to be identical with oxytocinase. However, sera from bovine contain only one kind of LAP activity—LAP3, which is different from the ones in human. Bovine LAP3 are ubiquitous in multiple organs and a role in catabolizing proteolytic degradation products has been proposed. Bovine LAP3 gene maps to chromosome 6 and encompasses 13 exons spanning a 25 kb genomic segment and encoding a 519 amino acid mature protein. Previously authors observed significant variations in LAP3 expression across different stages of the lactation cycle, with peak expression levels during lactation, reaching 2.2 times the levels observed during gestation, and confirmed variation of LAP3 gene to milk quality variation. Authors investigate the genetic polymorphism of the LAP3 gene and its association with yak milk quality traits.

Title is short and informative.

Abstract is adequate.

The Introduction is well written and covers the biological characteristics of Yak cattle and the importance of their milk. In the next paragraph, genotypic analyzes conducted so far in cattle for milk production are described, and the importance of the LAP3 gene is emphasized. In the end, the authors hypothesize that this gene may be significant in the prediction of milk quality.

MandM begins with an ethical statement and a description of the general experimental conditions. The next chapter describes DNA sampling and extraction. Please describe the procedures for Ear tissue sampling in more detail. Describe how milk is sampled (first stream, last stream, middle stream, batch). You can quote some protocols if they already exist. Was animal health surveillance carried out? How is mastitis treated? What other data was recorded: production and chemical composition of milk, stage of lactation, age and others. Was it measured from the same samples or from other samples during systematic herd monitoring? Genotyping and statistical processing are well described.

The results are clear and unambiguous, presented in textual, tabular and graphical form. In the results, the gene analysis of LSP3 is presented first. Gene frequencies and the existence of Hardy-Weinberg equilibrium were determined. The results are presented graphically through the Distribution of SNP genotypes, as well as tabularly where you can see all the calculated data. In the next step, correlation analysis of LAP3 genotype and dairy quality traits in Gannan yak was presented. In this part, the main results obtained are presented and the relationship of genes with protein status and other related milk properties is shown. In addition to relationships, the authors show that mutations appear to have a positive impact on milk quality in 227 Gannan yak, with mutant yak individuals exhibiting higher milk quality. Please explain the statement in line 243 in detail.

The discussion is well written and follows the results obtained. The conclusion follows from the above results.

References are adequate. What is “[J]” appearing in the literature?

Author Response

Reviewers' comments:

Reviewer #1:

  • MandM begins with an ethical statement and a description of the general experimental conditions. The next chapter describes DNA sampling and extraction. Please describe the procedures for Ear tissue sampling in more detail. Describe how milk is sampled (first stream, last stream, middle stream, batch). You can quote some protocols if they already exist. Was animal health surveillance carried out? How is mastitis treated? What other data was recorded: production and chemical composition of milk, stage of lactation, age and others. Was it measured from the same samples or from other samples during systematic herd monitoring? Genotyping and statistical processing are well described.

Reply: Thank you very much for your valuable comments, we have revised the manuscript according to your suggestions.

“In this study, 162 Gannan Yak cows from Xiahe County, Gannan Tibetan Auton-omous Prefecture, Gansu Province (altitude ranging from 3,000 to 3,800 meters) were selected as experimental subjects. All animals were in their lactation period and had calved 2 to 3 times. Samples were collected on the same day. The selected yak cows ex-hibited similar body conditions and comparable milk yields. These 162 Gannan Yak cows had been monitored by the local animal husbandry station over an extended pe-riod and were confirmed to be free from mastitis and reproductive-related diseases. They were grazing on the same natural pasture in the Gannan region and were man-aged under a fully pastoral system without any artificial feed supplementation.”

"During the experimental process, we also measured various milk components of the Gannan Yaks, such as density, citric acid, urea, and free fatty acids (FFA). However, since several loci of the LAP3 gene showed no association with these four milk quality traits, this data was not included in the article."

  • The results are clear and unambiguous, presented in textual, tabular and graphical form. In the results, the gene analysis of LAP3 is presented first. Gene frequencies and the existence of Hardy-Weinberg equilibrium were determined. The results are presented graphically through the Distribution of SNP genotypes, as well as tabularly where you can see all the calculated data. In the next step, correlation analysis of LAP3 genotype and dairy quality traits in Gannan yak was presented. In this part, the main results obtained are presented and the relationship of genes with protein status and other related milk properties is shown. In addition to relationships, the authors show that mutations appear to have a positive impact on milk quality in 227 Gannan yak, with mutant yak individuals exhibiting higher milk quality. Please explain the statement in line 243 in detail.

Reply: Thank you very much for your friendly suggestion. Regarding the content mentioned at line 243, we sought to preliminarily investigate the functional pathways enriched by the LAP3 gene. Therefore, we utilized the online tool KOBAS to identify the pathways enriched by the LAP3 gene. The results indicated that the LAP3 gene is enriched in pathways such as arginine and proline metabolism and glutathione metabolism, which influence milk quality.

  • References are adequate. What is “[J]” appearing in the literature?

Reply: Thank you very much for your valuable comments. Due to an oversight in our work, this issue has arisen. We have made the necessary corrections in the manuscript by removing '[J]' from the reference list.

Reviewer 2 Report

Comments and Suggestions for Authors

Abstract: no change needed.

Introduction:

In line 50, authors stated, “Additionally yak meat is rich in minerals and vitamins.” Aren’t all meats being like that?

The paragraph ends in line 87 stated about human and bovine SNPs. Then next paragraph suddenly jumped to LAP3 gene. Here if the authors add a line that yak is a bovid and they are looking for possible gene of interest for biomarkers. Also please mention why you choose LAP3. Did LAP3 came as gene of interest in previous studies? If yes then please add reference. Then authors can describe LAP3 gene.

Material and methods:

Authors mentioned about collecting ear tissues in line 121. If they are wild yak then I can understand about the danger collecting blood samples. But if they are farm yak then authors should have collected blood as DNA source. Can authors please explain why they did not collect blood samples.

In section “2.4: Genotyping”, there authors used cGPS chip for SNP detection. In this chip did they only analyze the gene of interest? Did they use any gene/genes as controls.

In section 2.5, milk composition was analyzed. Did the author present the data in result section? I did not see. It would be great if authors add a table to compare milk quality of yak with other bovids (cow, goat, camel).

Result:

In line 187-190, authors mentioned about four SNP of interest, but all of them are moderate to low polymorphism. Were there any SNP’s that showed high polymorphism? Why authors picked these 4 SNP’s?

In table:2 for g.15,61A>G why there was no GG genotype?

Discussion:

In line 256, “The Observer”, is this a newspaper?

In ine 259-260 authors stated protein content of yak milk is more than cow milk without any reference.

In line 294, authors mentioned about LAP3 genes function on limiting SARS-CoV2 inflammation. Authors may suggest it’s possible role on yak respiratory function since it lives in low oxygen environment.

Conclusion:

In future are the authors planning to develop a PCR based easy test to check these genotypes in larger yak population?

Author Response

Responds to the reviewer’s comments

Reviewers' comments:

Reviewer #2:

  • In line 50, authors stated, “Additionally yak meat is rich in minerals and vitamins.” Aren’t all meats being like that?

Reply: Thank you very much for your valuable comments, Due to our mistakes, this sentence is misdescribed. we have revised the manuscript according to your suggestions.

“The yak meat produced in the natural plateau pasture has its unique characteristics, its meat color is tender, bright red, and contains high protein, low fat, and rich minerals and vitamins.”

  • The paragraph ends in line 87 stated about human and bovine SNPs. Then next paragraph suddenly jumped to LAP3 gene. Here if the authors add a line that yak is a bovid and they are looking for possible gene of interest for biomarkers. Also please mention why you choose LAP3. Did LAP3 came as gene of interest in previous studies? If yes then please add reference. Then authors can describe LAP3 gene.

Reply: Thank you very much for your friendly suggestion. we have revised the manuscript according to your suggestions.“Yak milk production is influenced by many factors, and studies have shown that many genes (LAP3, CSN1S1, DGAT1 and RPL8) are related to milk production traits. The influence of these genes on the proteins in milk is well known”

  • Authors mentioned about collecting ear tissues in line 121. If they are wild yak then I can understand about the danger collecting blood samples. But if they are farm yak then authors should have collected blood as DNA source. Can authors please explain why they did not collect blood samples.

Reply: Thank you very much for your valuable comments. As you said, for wild yaks, collecting blood samples is difficult and dangerous. When we collected the samples of these 162 yaks, they were lactating. If we collected the blood samples of these yaks, we needed to drive and fix them artificially, which would greatly increase the stress on the female yaks. The process of collecting ear tissue for yaks is relatively less stressful and easy to operate. The location of the sample is more remote, and the DNA in the ear tissue sample is more stable than that in the blood sample. At the same time, ear tissue samples can provide sufficient amounts of high-quality DNA for genotyping.

  • In section “2.4: Genotyping”, there authors used cGPS chip for SNP detection. In this chip did they only analyze the gene of interest? Did they use any gene/genes as controls.

Reply: Thank you very much for your friendly suggestion. We used lllumina Yak cGPS 7K liquid phase chip to genotype 162 Gannan yaks. cGPS designs specific probes for target interval sequences based on an optimized thermodynamic stability algorithm model. Using synthetic specific probes, multiple different target sequences at different genomic locations were captured and enriched through liquid phase hybridization, and then library construction and second-generation sequencing were performed on the captured and enriched target regions, so as to obtain the genotypes of all SNP/InDel marker sites in the target regions. The genomic location of the SNP was derived from the assembly of the yak reference genome Bosgru v3.0[34] (GCA_005887515.1).

  • 在第 2.5 节中,分析了牛奶成分。作者是否在结果部分提供了数据?我没有看到。如果作者添加一个表格来比较牦牛与其他牛科动物(奶牛、山羊、骆驼)的牛奶质量,那就太好了。

Reply: Thank you very much for your valuable comments. Due to our mistake, we did not include this data in the results, because we mainly wanted to reflect the correlation between the four loci of LAP3 gene and dairy quality traits, so as to provide potential genetic molecular markers for molecular breeding of Gannan yak. It also provides data reference for improving yak milk quality and molecular markers of yak population. Therefore, in the section of correlation analysis between genes and dairy quality traits, the relevant data of genotypes at different loci are listed. No data was provided for the 162 Gannan yaks and no comparison was made with other species.

  • In line 187-190, authors mentioned about four SNP of interest, but all of them are moderate to low polymorphism. Were there any SNP’s that showed high polymorphism? Why authors picked these 4 SNP’s?In table:2 for g.15,61A>G why there was no GG genotype?

Reply: Thank you very much for your valuable comments. In the LAP3 gene loci we discovered, we did not find high polymorphism, but we found that among the four loci, one loci was in low polymorphism and three loci were in moderate polymorphism. As for the G.15,61A>G locus you mentioned, we only detected two genotypes. This may be because there are only two different alleles in a single locus, so two genotypes appear.

  • In line 256, “The Observer”, is this a newspaper?In ine 259-260 authors stated protein content of yak milk is more than cow milk without any reference.In line 294, authors mentioned about LAP3 genes function on limiting SARS-CoV2 inflammation. Authors may suggest it’s possible role on yak respiratory function since it lives in low oxygen environment.

Reply: Thank you very much for your friendly suggestion. As for The Observer, it was only mentioned in one of their reports. We added references in the manuscript to the part that the protein content of yak milk is higher than that of cow's milk. As for the description of the anti-infection effect of LAP3 gene on "SARS-CoV2", the description in the paper seems to be inaccurate and inappropriate, which is not consistent with the content of the paper, so we have deleted it.

  • 将来,作者是否计划开发一种基于 PCR 的简单测试来检查大型牦牛种群中的这些基因型?

Reply: Thank you very much for your valuable comments. We hope that the new SNPs in LAP3 gene can be used as potential genetic markers for genetic improvement in yak breeding programs and as molecular markers for dairy quality traits of yaks. In order to improve the accuracy, we tried to establish this simple PCR for the detection of yak population.
